behaviour/physiology

glucocorticoid metabolite, horse, immunoglobulin A, motor, sensory, laterality

**Authors for correspondence:**
I. Marr
e-mail: marri@stud.hfwu.de
K. Krueger
e-mail: konstanze.krueger@hfwu.de

# Non-invasive stress evaluation in domestic horses (*Equus caballus*): impact of housing conditions on sensory laterality and immunoglobulin A

I. Marr[1,2], V. Preisler[2], K. Farmer[3], V. Stefanski[2] and K. Krueger[1,4]

[1]Department Equine Economics, Faculty Agriculture, Economics and Management, Nuertingen-Geislingen University, Neckarsteige 6-10, Nuertingen 72622, Germany
[2]Behavioral Physiology of Livestock, University of Hohenheim, Garbenstr. 17, Stuttgart 70599, Germany
[3]School of Psychology & Neuroscience, University of St Andrews, St Andrews, Scotland KY16 9AJ, UK
[4]Zoology/Evolutionary Biology, University of Regensburg, Universitaetsstr. 31, Regensburg 93053, Germany

IM, 0000-0001-6272-0482; KK, 0000-0002-9120-680X

The study aimed to evaluate sensory laterality and concentration of faecal immunoglobulin A (IgA) as non-invasive measures of stress in horses by comparing them with the already established measures of motor laterality and faecal glucocorticoid metabolites (FGMs). Eleven three-year-old horses were exposed to known stressful situations (change of housing, initial training) to assess the two new parameters. Sensory laterality initially shifted significantly to the left and faecal FGMs were significantly increased on the change from group to individual housing and remained high through initial training. Motor laterality shifted significantly to the left after one week of individual stabling. Faecal IgA remained unchanged throughout the experiment. We therefore suggest that sensory laterality may be helpful in assessing acute stress in horses, especially on an individual level, as it proved to be an objective behavioural parameter that is easy to observe. Comparably, motor laterality may be helpful in assessing long-lasting stress. The results indicate that stress changes sensory laterality in horses, but further research is needed on a

larger sample to evaluate elevated chronic stress, as it was not clear whether the horses of the present study experienced compromised welfare, which it has been proposed may affect faecal IgA.

## 1. Introduction

It has been suggested that some horse management regimes, such as individual stabling without contact with conspecifics, may compromise the animal's natural needs and result in various types of stress responses, for example, increased stress hormone concentration, eye temperature and heart rate, and the display of stereotypic behaviour [1,2].

Stressors that last for a few minutes or hours may cause acute stress responses, while stressors that last for several hours per day over weeks or months may cause chronic stress responses [3]. Some consider acute stress responses to be 'positive', beneficial physiological responses and chronic stress responses to be 'negative', maladaptive responses, although the differentiation between them remains vague [4]. The duration and extent of a physiological stress response has been shown to depend on the individual's stress sensitivity, perception and processing of the stressor, in humans [5], rats [6,7], mice [8], pigs [9] and dogs [10]. Animals' stress levels are commonly assessed by analysing the levels of glucocorticoid metabolites (GMs) in different body fluids. Minimally to non-invasive methods, including the analysis of stress hormones such as glucocorticoids in saliva [11] or their metabolites in faeces [12], allow samples to be taken with little or no stress to the animals. Blood sampling for the measurement of stress hormones is invasive and may cause a stress response in wild, domestic and laboratory animals [13–17]. Furthermore, the analysis of glucocorticoids may not be suitable for differentiating between short-term and long-term stress because glucocorticoid production can be increased by both [5], and in the case of chronic compromised welfare, may decrease to baseline levels and below after an initial increase [18].

Acute stress situations, which activate the sympathetic adrenal medullary system, elevate the number and activity of certain immune cells, i.e. blood neutrophils, providing an enhanced first line immune defence [3,6,19]. Furthermore, cortisol is released when stress activates the hypothalamic-pituitary-adrenal axis causing a redistribution of lymphocytes from the blood to different organs for immune defence [3,6,19]. The number and activity of leucocytes may influence immunoglobulin A (IgA) production and concentration in various different ways, including the number and activity of B-lymphocytes, plasma cell activity and the production of secretory component [7,20]. Additionally, the intensity and duration of a stressor modulates the production of secretory IgA [21]. In many species, IgA is the most commonly secreted immunoglobulin in the gastrointestinal tract and has an important role in mucosal immune defence [22]. In general, it has been shown that secretory IgA increases under acute stress but decreases under chronic stress [23] in rats [7], mice [8], pigs [9] and horses [24]. Therefore, non-invasive IgA analysis may help to determine stress in animals reliably.

The analysis of physiological stress responses may be supplemented with observations of behaviour [18], but as the interpretation of animal behaviour is complex and time consuming, a simple, quickly determinable, objective behavioural parameter is needed. It has been proposed that lateral limb, paw, claw or hoof use, i.e. motor laterality [25–29], is such an indicator. As motor functions are controlled by the contralateral brain hemisphere [30], a left shift in limb use indicates increased processing by the right hemisphere, and a right shift indicates increased processing in the left hemisphere. Stress leads to increased information processing by the right brain hemisphere, which has been shown to control responses to stress, novelty, social interactions and predators [25]. The right hemisphere also controls the sympathetic nervous system activity, indicated by increased cardiac activity in horses under stress [31]. At the physiological level, increased information processing by the right hemisphere has been linked to changes in immune reactivity such as higher blood lymphocyte numbers, lower IFN-γ production and reduced antibody response [27,28], and has been shown in mice [32] and dogs [33,34]. The left hemisphere has been shown to be responsible for categorization of stimuli and routine situations [25,26,35–41].

As with motor laterality, sensory laterality is also associated with one-sided hemispheric information processing [42,43]. Social information, both in agonistic and affiliative contexts, is processed by the left hemisphere and this reflects the need to respond quickly and appropriately to emotional information [43]. The use of sensory organs on the left side has been shown to correspond with increased emotionality in horses [31,42,44–46], red-capped mangabeys [47], lizards [48] and dogs [49]. However,

it remains unclear how stress affects visual and auditory laterality. Sensory laterality has been said to be a flexible parameter [50], and so may change faster and be more situation related than motor laterality.

Horses are good model organisms for comparing the established stress indicators of GMs and motor laterality with the potential stress parameters of IgA and sensory laterality, because there has already been substantial work in this field. Previous studies have demonstrated that stressors such as changes from social housing to individual housing (which restricts movement and social interaction) increase the concentration of cortisol both in horses' saliva and of GMs in the faeces [1,2]. The initial training increased cortisol concentrations in horses' saliva [51] but not the GM concentrations in faeces [52]. It has also been demonstrated that stress associated with road transport increased, and physiological responses associated with anaesthesia decreased, faecal IgA concentration [24]. Motor laterality has been shown to be evenly distributed at population level under normal circumstances in free-roaming Przewalski and domestic horses [37,53]. However, it has been suggested that stress causes a population preference for left forelimb use in Quarter Horses [40], and it is possible that sensory laterality is also affected by stress. However, it has been proposed that enhanced lateral forelimb use [35,37,53] and enhanced sensory laterality [54] may partially be caused by one-sided training in horses. Any left or right preference in sensory organ use is easy to assess in horses, as horses' eyes and ears are laterally positioned, and they generally shift their heads when switching between the use of the sensory organs of one side and the other.

The current study aimed to evaluate acute and long-term stress responses in horses to answer the question of whether faecal IgA and sensory laterality change comparably with the established stress parameters of GMs and motor laterality in known stress situations as suggested by researchers [1,2,24,42,44,51]. It was conducted in three-year-old castrated male riding horses and addressed the change from social to individual housing and initial training, which are known to cause stress responses in horses, followed by regular training and continuing individual stabling for two months to analyse possible effects of such challenging situations.

# 2. Material and methods

## 2.1. Animals and location

Data were collected between November 2015 and January 2016 at the state stud farm Marbach, Gomadingen, Germany. The state stud farm provided 12 three-year-old (foaled in 2013), castrated male German Warmblood horses (*Equus caballus*) which were raised together in one social group. The horses were identified through their coat colours and white markings on their heads and legs. All animals were in good health and feeding condition and were checked by the stud's veterinarian on a regular basis. The horses were used to being led on a halter and being tied up to eat grain from wall mounted feeders.

At the start of the experiment, all 12 horses were kept in open housing at Marbach and had no previous experience of single boxes. They were kept in one group on a single 5.9-hectare pasture with a barn for shelter and handling. They had permanent access to grass and water, and received hay in addition. When they were in the barn for routine handling and veterinary treatment, they received about 500 grams of grain each. After they had been housed in a stable group composition at this facility for five weeks, they were led to the stabling complex where they were housed in 12 individual boxes (sized $3.2 \times 3.5$ m). The boxes allowed visual and physical contact between the horses, but only through the metal bars which separated the boxes. In the boxes, they received hay and water ad libitum, and grain three times a day (the amount depended on individual needs). Straw bedding was used in the barn and in the individual boxes. This experiment was restricted to 12 horses, as the stabling complex only had 12 suitable individual boxes.

## 2.2. Experimental procedure

The horses were exposed to four test situations (for an overview of the time schedule see electronic supplementary material, table S1):

### 2.2.1. A change from social housing on pasture to individual housing in single boxes

Test situation a: This test situation was expected to cause acute stress responses as a result of a changed environment, reduced physical contact to conspecifics and movement restriction [1,2]. Faecal GMs

(FGMs) were expected to be elevated, motor laterality to be unchanged, faecal IgA to be elevated and sensory laterality to be changed. During the first week, the horses received exercise in groups of six, in the form of free movement in an indoor arena, for 30 min per day. During the first week of individual housing, one horse was injured by another during free exercise and had to be excluded. Therefore, 11 horses were studied.

### 2.2.2. After one week of individual housing

Test situation b: After one week of individual stabling as described above, the horses were expected to show different coping strategies. Some horses might be quicker to adapt to the novel situation than others. Different stress responses were expected including still elevated FGMs, changed motor laterality, decreased faecal IgA (in the case of reactive coping strategies), FGMs at baseline level, unchanged motor laterality and baseline faecal IgA (in the case of proactive coping strategies). Sensory laterality was expected to return to baseline. The group average would depend on how many animals cope actively or passively.

### 2.2.3. Initial training

Test situation c: This commenced after one week of individual stabling and was expected to elicit acute stress responses again [51], with increased FGMs but unchanged motor laterality, as each training session lasted for less than 1 h [3]. The initial training consisted of 20 min lunging in an indoor arena on two consecutive days. Two horses were lunged in the arena at the same time to reduce isolation stress.

### 2.2.4. Regular training and individual stabling with no access to pasture or paddock

Test situation d: The horses were stabled in single boxes and received regular training (five times a week) for two months, as is the usual practice in Germany (most horses are stabled in single boxes, and training practice is in accordance with the guidelines of the German Equestrian Federation (FN)) [55]. This situation was assessed for possible long-term influences on the investigated parameters. It was possible that these parameters could still be slightly altered, depending on the individuals' coping strategies (FGMs elevated or reduced in the case of compromised welfare, motor laterality changed, IgA reduced in the case of compromised welfare and sensory laterality changed).

## 2.3. Sampling and behavioural observation

GM and IgA analyses, as well as observations of motor laterality and sensory laterality, were conducted in all test situations. Faecal samples for the baseline values were collected in the open barn while the horses were tied up for feeding, and samples for the test situations were collected from the individual boxes. All horses served as their own controls.

To calculate stable mean baseline values for each horse, three samples were taken for FGM and faecal IgA 7 days before, 6 days before, and 1 day before the first test situation (a). As horses' intestinal passage takes about 24 h [56], faecal samples were taken during each test situation as follows: (a) 24 h and 48 h after the change of housing conditions, (b) one week after the change of housing conditions, (c) 24 h and 48 h after the first training session, (d) after two months of individual stabling (for an overview on the sampling schedule see electronic supplementary material, table S1). Motor and sensory laterality were observed once to establish baseline values and for each test situation with observations spread over three consecutive days, as detailed below in §2.7.

## 2.4. Glucocorticoid metabolites

To determine GMs, faecal samples were taken between 8.00 and 10.00 to control for diurnal variations (IgA [57], GM [58]). Fresh faecal samples were collected from the centre of the pile with a freezer bag between 1 and 2 min after defecation and kneaded for 1–2 min to ensure an equal distribution of IgA and GMs. They were then kept on ice in a cool box until they were frozen in the laboratory at −20°C to avoid a decline in FGM concentration through bacterial decay, as has been demonstrated elsewhere [59]. The samples were processed as described elsewhere [60]. GMs were extracted from faecal samples by adding 5 ml of 80% methanol [61] to 0.5 g wet faeces. The suspension was vortexed for 2 min, incubated at room temperature for 15 min and vortexed again. After the centrifugation at 2500$g$

for 15 min, the supernatant was aliquoted and frozen at −20°C until further analysis [62]. As validated and described for horses elsewhere [60], the diluted supernatant (in assay buffer) was analysed by an 11-oxoaeticholanolone enzyme immunoassay measuring 11,17-dioxoandrostanes.

Several enzyme immunoassay plates were needed. Therefore, samples at the plates were pipetted always in the same order, starting with horse number 1. In addition, samples from the same two control values were used on all plates, always at the end and in the same wells to calculate an inter-assay coefficient of variation (CV). Control 1 had a CV of 0.18 and control 2 of 0.20, which is slightly above the elsewhere described CVs [60] which were between 0.13 and 0.16. To allow statistical comparison between samples from different plates, all samples were multiplied by a correction factor, which was calculated as follows for each plate: the mean value of the control samples from all plates divided by the actual control value of each plate.

## 2.5. Immunoglobulin A

Faecal samples were collected as described for GMs in §2.4. The protocol for the extraction of IgA from faecal samples was adapted from elsewhere described protocols [57,63–65]. Faecal samples were thawed at room temperature (21°C). Ten millilitres of phosphate-buffered saline were added to 1 g wet faeces in a tube. The suspensions were shaken (with the top of the tube downwards), vortexed for 3 min and incubated for 15 min at room temperature. This process was then repeated. The samples were then centrifuged for 20 min at $1600g$. The supernatant (1.2 ml) of the suspension was transferred into a new tube and centrifuged for 15 min at $3260g$ (21°C). The supernatant was aliquoted and frozen at −20°C until further analysis. IgA concentration was measured by enzyme immune assay (EIA) according to the manufacturer's protocol of the Horse IgA enzyme-linked immunosorbent assay (ELISA) Quantification Set (E70-116, Bethyl Laboratories, Inc., Montgomery, USA).

As for GM analysis, on all plates, the same two control values were used to calculate an inter-assay CV for IgA. The CV for IgA was 0.12 and 0.13. A correction factor was used as described for GMs.

## 2.6. Motor laterality

Motor laterality was observed by scan sampling, recording how often the left or right forelimb of the horse was placed in front while grazing on pasture and while eating hay from the floor in the box [37]. Sixty observations were made at 30 s intervals, at different times of day, spread over up to 3 days, for each horse in each test situation.

A motor laterality index (MLI) was calculated for each horse as described elsewhere [37]: $LI = (R − L)/(R + L)$, where R describes the number of observations when the right forelimb was in front and L describes the number of observations when the left forelimb was in front. A negative MLI indicates a left preference and a positive MLI indicates a right preference.

## 2.7. Sensory laterality

Sensory laterality was observed using ad libitum sampling. When a horse raised its head to observe the environment, the direction in which the head was turned was recorded if it was 30 degrees or more to the left or right, as described elsewhere [53]. The observations were made for 2 h for each test situation. Additionally, sensory laterality was assessed using a novel-object test which was conducted either on the pasture or in the individual boxes, according to the test situation. On the pasture, the test horse was separated from the others with the help of a person who prevented the other horses from approaching. In the stabled situations, the novel-object test was done in the horses' boxes. In both cases, the novel-object was placed 1–2 m in front of each horse and we recorded which side of the head (left, right or frontal) was used for the initial investigation of the object. In each situation, six unfamiliar objects from the following list were presented to each horse: cones of various colours, balls of various colours, a rubber boot, buckets in different colours, an unfilled air mattress, a blanket, pieces of pipes in different colours, cartons of various shapes and colours, half a swimming noodle and bags of various colours filled with straw.

A sensory laterality index (SLI) was calculated for each horse as described for motor laterality in §2.6.

## 2.8. Experimenters

The two experimenters were PhD and Master students from Nuertingen-Geislingen University and University of Hohenheim. Before the experiment started, the inter-observer agreement between the two experimenters was tested for the assessment of sensory and motor laterality and a Cohen's Kappa coefficient was calculated (sensory laterality: $\kappa = 0.75$, motor laterality: $\kappa = 0.93$). Behavioural parameters (sensory laterality, motor laterality) were recorded on paper. One experimenter collected and kneaded the faecal samples, while the other observed the motor laterality and the head lifts for the sensory laterality. Both experimenters conducted the novel-object test to determine sensory laterality. The horses were handled and trained by employees of the state stud farm Marbach.

## 2.9. Statistical analysis

The R Studio (v. 0.99.484, Boston MA, USA) and R commander (2.2-1) were used for the statistical analysis. Figures were constructed with BP-tool (Add-In) by Microsoft Excel 2010 (Microsoft Corporation, Washington, USA). Three baseline measurements were taken for GM and IgA each to calculate stable mean baseline values for each horse during group housing (without experimental stressors; for all data see electronic supplementary material, table S2). The baseline values served as controls for comparison with values from the different test situations for each individual horse. As data were not normally distributed at all (Shapiro–Wilk test W < 0.83, N = 11, $p < 0.03$ for FGM 48 h training, IgA 24 h, 48 h, one week individual stabling, IgA 24 h, 48 h after initial training, motor laterality basal values, sensory laterality after initial training; for all others W > 0.86, N = 11, $p > 0.07$), we applied a generalized linear model (GLM) to analyse whether deviations between the baseline values and the test situations differed from 0 (formula = parameter [difference between baseline and test situation for each horse]~test situations, family = Gaussian (identity)). All tests were two tailed. The significance level was set at 0.05.

# 3. Results

## 3.1. Change from social to individual stabling

FGMs increased significantly from the baseline value taken at group housing to the value taken 24 h by 22.9 ng g$^{-1}$ (figure 1, GLM: FGM.difference ~ test situation, N = 11, $t = 3.5$, $p = 0.001$) and 48 h by 15.6 ng g$^{-1}$ (figure 1, GLM: FGM.difference ~ test situation, N = 11, $t = 2.4$, $p = 0.02$) after the change to individual stabling. IgA and motor laterality (ML) remained unchanged (figure 1, GLM: IgA.difference ~ test situation, N = 11, both $p > 0.05$; figure 2, GLM: ML.difference ~ test situation, N = 11, $t = -1.1$, $p = 0.26$). Sensory laterality (SL) shifted significantly to the left by LI −0.46 (figure 2, GLM: SL.difference ~ test situation, N = 11, $t = -2.4$, $p = 0.02$).

## 3.2. One week of individual stabling

After one week of individual stabling, FGMs returned to baseline values (figure 1, GLM: FGM.difference ~ test situation, N = 11, $t = 0.9$, $p = 0.38$) and IgA remained unchanged (figure 1, GLM: IgA.difference ~ test situation, N = 11, $t = -0.04$, $p = 0.97$). Motor laterality significantly shifted to the left by LI −0.25 after one week of individual stabling compared with baseline values (figure 2, GLM: ML.difference ~ test situation, N = 11, $t = -2.5$, $p = 0.01$). Sensory laterality showed a trend to shift to the left by LI −0.33 (figure 2, GLM: SL.difference ~ test situation, N = 11, $t = -1.7$, $p = 0.096$).

## 3.3. Initial training

FGMs showed a trend to be elevated 24 h after the first training session (initial training) by 12.1 ng g$^{-1}$ (figure 1, GLM: FGM.difference ~ test situation, N = 11, $t = 1.8$, $p = 0.07$), and 48 h after the first training session FGMs had significantly increased by 14.3 ng g$^{-1}$ (figure 1, GLM: FGM.difference ~ test situation, N = 11, $t = 2.2$, $p = 0.03$). IgA remained unchanged (figure 1, GLM: IgA.difference ~ test situation, N = 11, both $p > 0.05$). Motor laterality and sensory laterality returned to baseline values after the first training session (figure 2, GLM: ML.difference ~ test situation, N = 11, $t = -1.6$, $p = 0.11$; GLM: SL.difference ~ test situation, N = 11, $t = -1.6$, $p = 0.13$).

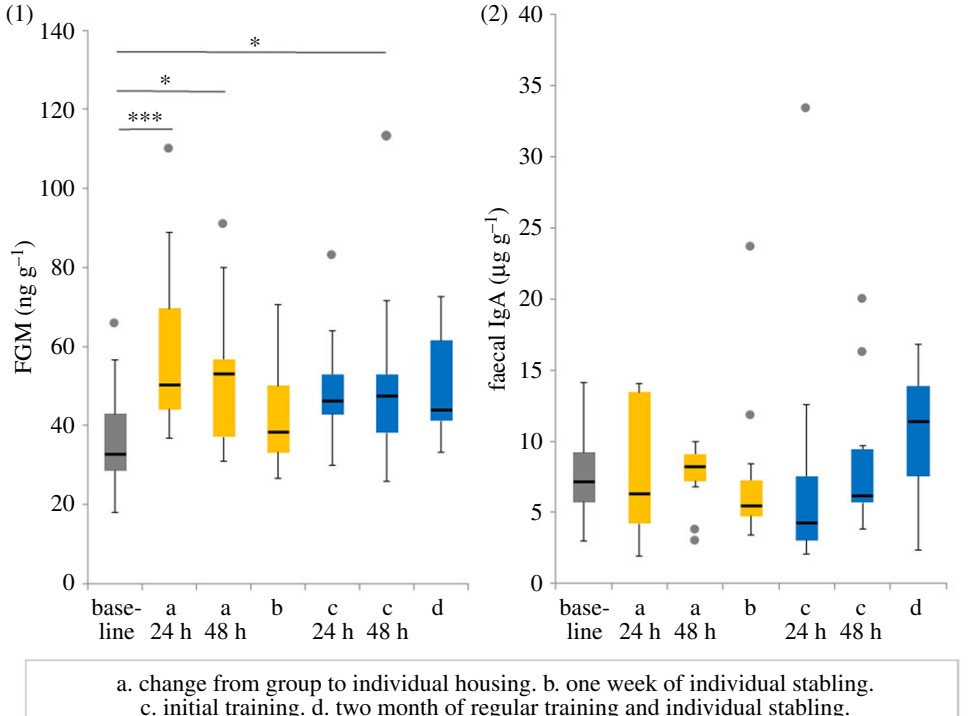

a. change from group to individual housing. b. one week of individual stabling.
c. initial training. d. two month of regular training and individual stabling.

**Figure 1.** Physiological and immunological stress parameters. FGM concentrations (1) and faecal immunoglobulin A concentrations (2) in test situation without stressors (base, during group housing) and 24 h (a 24 h) and 48 h (a 48 h) after the change of housing condition, one week of changed housing condition (b), 24 h (c 24 h) and 48 h (c 48 h) after initial training, and two months of regular training and individual stabling (d). Yellow: changed housing conditions, blue: combination of individual housing and initial/regular training. Box plots display the medians, interquartile ranges from 25% to 75%, whiskers (minimum and maximum values) and outliers (dots) for values higher or lower than 1.5 interquartile range. Outlier at 89 $\mu$g g$^{-1}$ faecal IgA 24 h after the change from group to individual housing is not shown by the figure. $^{***}p < 0.001$, $^{*}p < 0.05$.

## 3.4. Two months of regular training and individual stabling

Compared with baseline values FGMs tended to increase by 13.1 ng g$^{-1}$ (figure 1, GLM: FGM.difference ~ test situation, $N = 11$, $t = 2.0$, $p = 0.05$) after two months of individual stabling and regular training. IgA remained unchanged (figure 1, GLM: IgA.difference ~ test situation, $N = 11$, $t = 0.6$ $p = 0.54$). Motor laterality significantly shifted to the left by LI $-0.24$ (figure 2, GLM: ML.difference ~ test situation, $N = 11$, $t = -2.4$, $p = 0.02$). Sensory laterality remained unchanged (figure 2, GLM: SL.difference ~ test situation, $N = 11$, $t = -1.5$, $p = 0.14$).

# 4. Discussion

The change from group to individual housing and the combination of individual housing with initial/regular training caused related stress responses indicated by changes in FGMs and motor laterality. After two months, FGMs continued to show an elevated trend, and the motor laterality changes persisted. Significant left shifts in sensory laterality occurred in parallel with increases in the established stress parameter FGMs, and this provides more insight into acute stress responses resulting from changed housing conditions. IgA concentrations did not change significantly. The change from group to individual housing is considered to be a long-term stressor [1,2]. In the present study, significantly increased FGMs suggested an acute stress response that declined after one week. Afterwards FGMs tended to remain elevated compared with baseline. Therefore, it is presumed that the horses did not experience severely and permanently compromised welfare which may reduce GMs as described elsewhere [18]. In addition, motor laterality shifted to the left as predicted by other researchers [36]. Prior to the initial training the observed left shift in the present study was not caused by one-sided training [35,37,53], as the horses were already accustomed to being handled from the left in their previous social housing. Therefore, the results support the proposal that motor laterality is a suitable

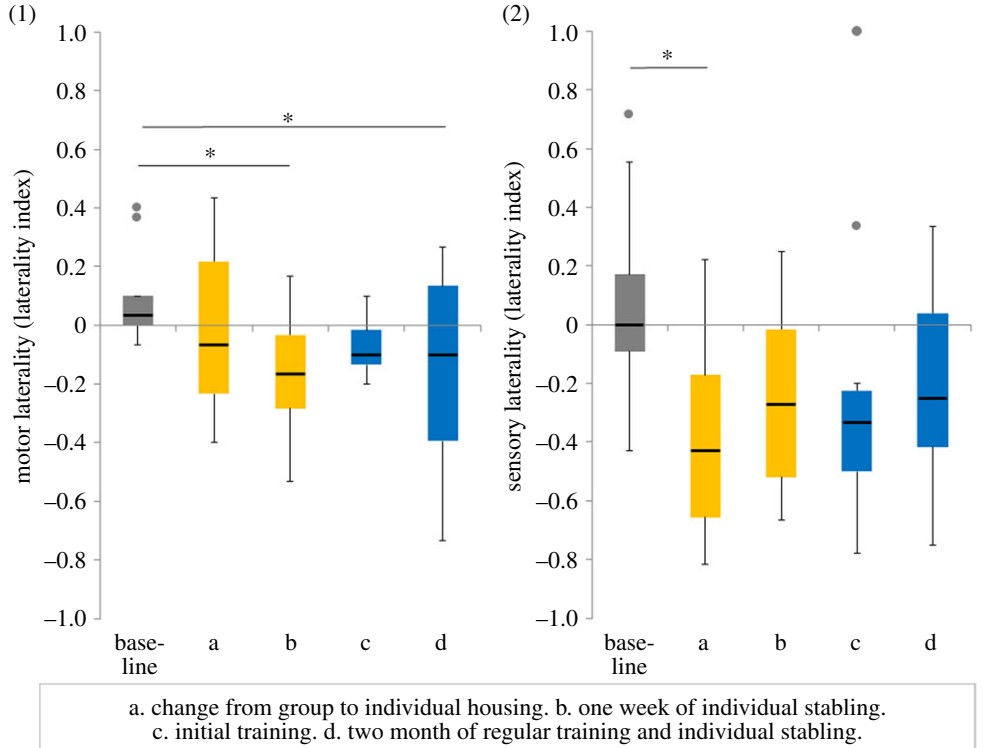

a. change from group to individual housing. b. one week of individual stabling.
c. initial training. d. two month of regular training and individual stabling.

**Figure 2.** Motor and sensory laterality indices. Motor laterality (1) and sensory laterality (2) in test situation without stressors (base, during group housing) and after the change of housing condition (a), one week of changed housing condition (b), after initial training (c) and two months of regular training and individual stabling (d). Yellow: changed housing conditions, blue: combination of individual housing and initial/regular training. Box plots display the medians, interquartile ranges from 25% to 75%, whiskers (minimum and maximum values) and outliers (dots) for values higher or lower than 1.5 interquartile range. $^*p < 0.05$.

behavioural parameter to supplement physiological stress assessment for long-lasting stress conditions [25,26].

Sensory laterality may also be a suitable behavioural parameter for assessing acute stress responses as it shifted significantly to the left with the change from group to individual housing. As a left shift in sensory and motor laterality was found in parallel with the increase of FGMs, and as the right brain hemisphere controls cortisol secretion in emotional situations [66], it is suggested that the dominance of the right hemisphere in the observed stressful situations caused the left shift in sensory and motor laterality [25,26].

As long-term stress has been repeatedly reported to have immuno-suppressive effects [3,19,21,67], a decreased IgA concentration was expected. However, IgA concentrations remained unchanged. As indicated by the other stress parameters, the stress in the specific test situations may not have been strong and long enough to trigger decreases in faecal IgA concentration [8,21]. Obviously, further research is required, especially on the effect of chronic stress on IgA in horses.

All the parameters showed large inter-individual variation. This may have been due to individual stress sensitivity, perception and the processing of the stressor [3,5,6]. For some horses, the ongoing training and the novel housing may have become routine more quickly and led to a decline in stress responses [68]. Therefore, further research on individual responses is required.

This study shows that the investigated acute and long-lasting challenging situations caused changes in the investigated stress parameters. Sensory laterality appears to be a good behavioural parameter for the non-invasive evaluation of acute stress responses, such as the change from social to individual housing, that involve a change in environment as well as a restriction in movement and reduced contact with conspecifics. Sensory laterality changed more quickly and was more situation related than motor laterality. However, it remains unclear whether the left shift in motor laterality in these maturing horses would persist in continued individual housing and training, as the experimental period lasted only for two months. This requires further research. Besides implicating animal welfare

issues [25], a left shift in sensory and motor laterality indicating an increased information processing by the right hemisphere could indicate training and handling issues. A higher left eye preference in animals has been reported to show enhanced emotionality or increased fearful behaviour [36,44], and so may indicate an increased likelihood of unpredictable and dangerous reactions during handling by humans. Furthermore, a significant increase in the horse's emotionality would have disadvantages for training, as emotionality has been shown to be negatively correlated with trainability in horses [69]. Apart from this, it has been demonstrated that left-sided horses are more likely to treat an ambiguous stimulus as negative [29]. While low or moderate stress can enhance trainability, prolonged and/or elevated stress hormone concentration can cause memory disruption [70]. Moreover, less stressed, pasture-kept horses have been shown to reach training criteria more quickly than stabled horses [71] and to be easier to handle and to train [72]. In addition, stress may lead to a higher prevalence of abnormal behaviour, stereotypies and depressive-type behaviour [73,74]. Therefore, the easy and objective evaluation of sensory and motor laterality may help to improve animal welfare, and future research on laterality and stress should focus on stress levels, degree of left shifts [75], emotionality, mental health and consequences for handling and training. In stress situations similar to those in the present study behavioural parameters have been shown to be potentially more indicative in situations where stress hormones did not show any change, for example, after changed housing conditions [76]. Nonetheless, in addition to changes in laterality, the evaluation of other stress parameters is also recommended for a reliable assessment of stress responses [77]. Although the sample size of our study is low, based on the results it is expected that the observation of left shifts in motor and sensory laterality may be helpful objective parameters for stress analysis in horses, especially on an individual level.

Faecal IgA was expected to be a sensitive, non-invasive parameter for detecting activity of sympathetic adrenal medullary system as a first step response to unpredictable mild acute stressors. However, in the comparatively more challenging and long-lasting test situations (b and d), IgA concentration may have been a result of downregulated adaptive immune function. In follow-up studies, stronger and more isolated stressors are needed to analyse whether chronic stress results in faecal IgA suppression.

# 5. Conclusion

Although further research is needed to fully understand the relationship between the investigated parameters, this study opens up new non-invasive stress parameters for evaluating animal welfare. Sensory laterality is a promising non-invasive parameter that may help to avoid exposing animals to additional stress through invasive sampling procedures. This study indicates that potentially stressful situations change sensory laterality in horses. Further research on a larger sample size is needed to evaluate chronic stress at greater intensities. Other studies have demonstrated that individuals differ in their coping strategies and ability to adapt to stress in novel and/or unnatural situations. The non-invasive parameters used in this study may allow animal welfare to be evaluated on an individual level, and motor and sensory laterality are easy to assess. Laterality may therefore be a promising parameter to help lay persons to identify stress in their horses, eliminate or reduce the cause and improve the horses' welfare.

Ethics. This study was carried out in strict accordance with the recommendations in the German animal welfare law. The study was approved by the responsible animal welfare board in Tuebingen, Germany (permit no. 35/9185.81-4/).

Data accessibility. Original data is available in electronic supplementary material: electronic supplementary material, table S2.

Authors' contributions. I.M. and V.P. collected the data; I.M., V.P., V.S. and K.K. substantial contributions to conception and design; I.M. and K.K. analysed data; I.M. written original draft preparation; K.F., V.S. and K.K. written review and editing; all authors agreed for publication and all agreed to be held accountable for all aspects of the work.

Competing interests. The author and co-authors of this manuscript have no competing interest.

Funding. This work was supported by 'Ministerium für Laendlichen Raum und Verbraucherschutz Baden-Wuerttemberg' (grant no. 8284.02) and Deutsche Reiterliche Vereinigung e.V. (FN).

Acknowledgements. We would like to thank the State Stud Farm Marbach especially Dr Astrid von Velsen-Zerweck, Dr Carolin Eiberger, Dr Yvonne Zander, Dr Sonja Schmucker for helping with the design of the study; Andrea Dobler for helping and analysis of faecal samples; two anonymous referees and the editor for helpful suggestions and the involved employees for cooperation and their time and efforts during the experiment.

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
