## [Reviewer comments · Royal Society Open Science]

Review History

RSOS-191994.R0 (Original submission)

Review form: Reviewer 1

Is the manuscript scientifically sound in its present form?

Yes

Are the interpretations and conclusions justified by the results?

Yes

Is the language acceptable?

Yes

Do you have any ethical concerns with this paper?

No

Have you any concerns about statistical analyses in this paper?

No

Recommendation?

Accept with minor revision (please list in comments)

Comments to the Author(s)

This presents a novel and useful study looking at non-invasive measures of both short-term and long-term stress in horses. This paper is well written, and only requires some small edits to enhance its readability.

Page 2

L34 - insert the word "initially" before "significantly shifted to the left..."

L35 - insert "and remained high through initial training" after "individual stabling."

I know you are probably limited by word count, but I feel it would be useful to include the results of your IgA in the summary.

L45 - change wording "...individual stabling without contact with conspecifics..."

Page 3

L24 - reference 31 deals with differences in right and left hemisphere processing, but in humans, not horses. Please adjust where this reference appears.

L44 - Change to read "...in free-roaming Przewalski and domestic horses.."

L48 - one-sided

L57 - two months (plural)

Page 4

L27-29 - Maybe move the description of the box stalls to the above section 3.1

L35-36 - Can you explain more clearly what responses you expected - ie. what was your hypothesis?

L47 - Please explain this in more detail as "usual practice" will differ among individuals, countries, etc.

L58 - please rewrite as either "To calculate stable mean base values..." or "To calculate a stable mean base value..."

Page 5

L11 - Please indicate the length of time for centrifugation.

L32,33 - Please write out the acronyms EIA and ELISA.

Page 6

L17 - Please explain what the "cs-tool" is.

L21 - Include the test statistic and p-value for the Shapiro Wilks test.

L47-49 - There is no mention of salivary cortisol analyses in the methodology. Furthermore, Fig 1 shows only values at 24h and 48h after initial training (not immediately or 2h after; nor are these time frames referred to in methodology). Also do not begin the next sentence with a numeral.

Page 7

L17-18 - could it also be that training further developed the "sidedness" of particular horses?

L41 - months (plural)

L56-57 - However, Squibb et al. (2018) showed that behaviour did not reflect internal shifts in heart rate variability or eye temperature. Perhaps this is due to mature horses used in that study (thus trained to suppress conflict behaviour) versus the untrained horses in this study. Squibb, K., Griffin, K., Favier, R., & Ijichi, C. (2018). Poker Face: Discrepancies in behaviour and affective states in horses during stressful handling procedures. *Applied Animal Behaviour Science*, 202, 34-38. doi: 10.1016/j.applanim.2018.02.003

Review form: Reviewer 2 (Rupert Palme)

Is the manuscript scientifically sound in its present form?

Yes

Are the interpretations and conclusions justified by the results?

Yes

Is the language acceptable?

Yes

Do you have any ethical concerns with this paper?

No

Have you any concerns about statistical analyses in this paper?

No

Recommendation?

Accept with minor revision (please list in comments)

Comments to the Author(s)

The paper submitted by Marr et al (RSOS-191994) reports a “stress” experiment (change of housing; initial training) in horses. The study aims at evaluating two new stress measures (sensory laterality and faecal IgA) by comparing them with two well established ones (FGMs and motor laterality). The latter clearly demonstrated the stressfulness of the situations, which was also found when using sensory laterality (but not faecal IgA). I think the experiment is well planned, the findings are interesting and, although preliminary, the suggested parameters (especially sensory laterality) are worth to be further investigated. Nevertheless there are several things, which to my opinion need to be revised and/or addressed adequately (see my detailed comments below) before the paper is suited for publication.

Detailed comments (ordered by line):

Page 2, line 32 (and elsewhere) I suggest adding the “s” for the plural: FGMs – in some cases the singular is needed (e.g. faecal glucocorticoid metabolite concentrations – FGM)

Line 37: modify to “level, as a behavioural parameter that....(I may be mistaken, but think that observing behaviour is always non-invasive, or?)

Abstract: I miss a word about unaffected IgA levels.

Line 46: stress responses (plural)

Line 49: better use “stressors” instead of “stress” (two times)

Line 57: Minimal to non-invasive methods ,..., such as glucocorticoids in saliva or their metabolites in faeces... (what is a gentle method?)

Page 3: line 0 and 1: GCs (or GC) instead of GM, because that is not only true for faeces, but a general statement.

Line 39: But see Gorgasser et al (2007), where no effect was seen. I suggest to include that ref (here and/or in the discussion), which indicated that it may depend upon breed or method.

Line 41: Ref 53 is not about FGMs – I suggest to replace it with a most recent review here (Palme, 2019).

Line 42: stress (no plural) or stressors. “decreased/increased” (Past Tense)

Page 4: line 4: why two and three-year-old? You only mention three-year-old elsewhere?

Line 16: “ad libitum” in italics

3.2. – a) and b): I suggest to combine those – there is nothing different in b) – just a few days later. It will also make the figs etc easier to comprehend.

It may be helpful to point to Table S1 already here (or even include a graph in the main text illustrating the timeline of the different situations and sample collections).

Line 35: “quicker”

3.3. – How fresh were collected samples? I think that’s a critical issue, as a recent study by the same research group (why not cite it?) demonstrated.

Line 55 (and elsewhere): I suggest to use “baseline” (adjective) instead of base throughout the ms!

Line 58: I wonder whether those 3 values were normally distributed, and think it may be better to use the median (more representative if there are outliers – and almost identical if not).

Maybe good to mention that the delay time of faecal excretion is about 1 day in horses, and therefore those sampling points were chosen.

Page 5, line 12: The EIA is not directed against 11-oxo-aetiocholanolone-17-CMO (the antibody was raised against that immunogen coupled to BSA). It's ok just to mention the name of the EIA (maybe include the group of FGMs picked up by this assay).

Line 19-21 (also lines 34 and 35): That correction sounds odd to me. What were intra-assay CV? If those were also large, you cannot "correct" with such a factor. By the way, I don't understand (from the wording) how it was calculated.

Line 27: delete "and mixed in the freezer bag by hand". You mention this already before – so why another mixing here?

Line 30 and 31: there should be a word space between a number and its dimension (1600 g; 1.2 ml; 3260 g)

Page 6; 3.8. I wonder why inter-observer correlation was calculated for motor laterality (line 8), if only one one did the observations (line 10).

4. a) and onwards. What are the given increases? Taken from mean or median values? Or calculated on an individual basis and from those values the mean/median?

Line 47: I was surprised to read about salivary cortisol here? At least I don't find it in Fig. 1 (and elsewhere).

Page 7, line 36: Reword: ...to be a good behavioural parameter for the non-invasive evaluation....

Line 41/42: The sentence is hard to understand, please reword.

Line 58: The sample size is always limited, but you probably mean "low".

References: Please carefully revise them. There are several refs with lacking full information (such as article number, issue or pages; e.g. 21; 29; 59,...)

Ref 24: Delete "Text"

Ref 29: I guess the journal is "Animals" issue/article number?

Ref 53: That is unsuited here – please replace by Palme, 2019.

Fig 1: As mentioned before: please modify "base" to baseline; delete the (b) – as the box is also in yellow you clearly indicate that it is the same situation (maybe rewrite to 1 d, 2 d, 7 d)

Delete the "concentrations" from both y-axes (that's trivial if you give the dimension).

Any explanation for the extremely high outlier in 2.? I suggest only mentioning it, but redrawing the figure with the y-axes scaled from 0 to 40!

Delete the "trend" (*). It is non-significant, and just makes the figure more complicated.

Fig 2: delete (*) – what is LI? The legends need to be self-explanatory.

References cited above:

Gorgasser, I., Tichy, A., Palme, R. (2007): Faecal cortisol metabolites in Quarter horses during initial training under field conditions. *Wien. Tierärztl. Mschr. - Vet. Med. Austria* 94, 226-230.

Palme, R. (2019): Non-invasive measurement of glucocorticoids: advances and problems. *Physiol. Beh.* 199, 229-243.

Decision letter (RSOS-191994.R0)

02-Jan-2020

Dear Ms Marr,

The editors assigned to your paper ("Non-invasive stress evaluation in domestic horses (*Equus caballus*): Impact of housing conditions on sensory laterality and immunoglobulin A") have now received comments from reviewers. We would like you to revise your paper in accordance with the referee and Associate Editor suggestions which can be found below (not including confidential reports to the Editor). Please note this decision does not guarantee eventual acceptance.

Please submit a copy of your revised paper before 25-Jan-2020. Please note that the revision deadline will expire at 00.00am on this date. If we do not hear from you within this time then it

will be assumed that the paper has been withdrawn. In exceptional circumstances, extensions may be possible if agreed with the Editorial Office in advance. We do not allow multiple rounds of revision so we urge you to make every effort to fully address all of the comments at this stage. If deemed necessary by the Editors, your manuscript will be sent back to one or more of the original reviewers for assessment. If the original reviewers are not available, we may invite new reviewers.

- Data accessibility

If you wish to submit your supporting data or code to Dryad (<http://datadryad.org/>), or modify your current submission to dryad, please use the following link:
<http://datadryad.org/submit?journalID=RSOS&manu=RSOS-191994>

- Competing interests

- Authors' contributions

- Acknowledgements

- Funding statement

Kind regards,
Andrew Dunn
Senior Publishing Editor
Royal Society Open Science
openscience@royalsociety.org

on behalf of Dr Claudia Wascher (Associate Editor) and Kevin Padian (Subject Editor)
openscience@royalsociety.org

Associate Editor's comments (Dr Claudia Wascher):

The presented paper investigates sensory laterality and concentration of faecal immunoglobulin as two potential non-invasive measures of stress in horses. Both reviewers find the study interesting and well presented and they have a number of suggestions to further improve the readability of the manuscript prior to publication.

Reviewers' Comments to Author:

Reviewer: 1

Comments to the Author(s)

This presents a novel and useful study looking at non-invasive measures of both short-term and long-term stress in horses. This paper is well written, and only requires some small edits to enhance its readability.

Page 2

L34 - insert the word "initially" before "significantly shifted to the left..."

L35 - insert "and remained high through initial training" after "individual stabling."

I know you are probably limited by word count, but I feel it would be useful to include the results of your IgA in the summary.

L45 - change wording "...individual stabling without contact with conspecifics..."

Page 3

L24 - reference 31 deals with differences in right and left hemisphere processing, but in humans, not horses. Please adjust where this reference appears.

L44 - Change to read "...in free-roaming Przewalski and domestic horses.."

L48 - one-sided

L57 - two months (plural)

Page 4

L27-29 - Maybe move the description of the box stalls to the above section 3.1

L35-36 - Can you explain more clearly what responses you expected - ie. what was your hypothesis?

L47 - Please explain this in more detail as "usual practice" will differ among individuals, countries, etc.

L58 - please rewrite as either "To calculate stable mean base values..." or "To calculate a stable mean base value..."

Page 5

L11 - Please indicate the length of time for centrifugation.

L32,33 - Please write out the acronyms EIA and ELISA.

Page 6

L17 - Please explain what the "cs-tool" is.

L21 - Include the test statistic and p-value for the Shapiro Wilks test.

L47-49 - There is no mention of salivary cortisol analyses in the methodology. Furthermore, Fig 1 shows only values at 24h and 48h after initial training (not immediately or 2h after; nor are these time frames referred to in methodology). Also do not begin the next sentence with a numeral.

Page 7

L17-18 - could it also be that training further developed the "sidedness" of particular horses?

L41 - months (plural)

L56-57 - However, Squibb et al. (2018) showed that behaviour did not reflect internal shifts in heart rate variability or eye temperature. Perhaps this is due to mature horses used in that study (thus trained to suppress conflict behaviour) versus the untrained horses in this study. Squibb, K., Griffin, K., Favier, R., & Ijichi, C. (2018). Poker Face: Discrepancies in behaviour and affective states in horses during stressful handling procedures. *Applied Animal Behaviour Science*, 202, 34-38. doi: 10.1016/j.applanim.2018.02.003

Reviewer: 2

Comments to the Author(s)

The paper submitted by Marr et al (RSOS-191994) reports a "stress" experiment (change of housing; initial training) in horses. The study aims at evaluating two new stress measures (sensory laterality and faecal IgA) by comparing them with two well established ones (FGMs and motor laterality). The latter clearly demonstrated the stressfulness of the situations, which was also found when using sensory laterality (but not faecal IgA). I think the experiment is well planned, the findings are interesting and, although preliminary, the suggested parameters (especially sensory laterality) are worth to be further investigated. Nevertheless there are several things, which to my opinion need to be revised and/or addressed adequately (see my detailed comments below) before the paper is suited for publication.

Detailed comments (ordered by line):

Page 2, line 32 (and elsewhere) I suggest adding the "s" for the plural: FGMs - in some cases the singular is needed (e.g. faecal glucocorticoid metabolite concentrations - FGM)

Line 37: modify to "level, as a behavioural parameter that...(I may be mistaken, but think that observing behaviour is always non-invasive, or?)

Abstract: I miss a word about unaffected IgA levels.

Line 46: stress responses (plural)

Line 49: better use "stressors" instead of "stress" (two times)

Line 57: Minimal to non-invasive methods ,..., such as glucocorticoids in saliva or their metabolites in faeces... (what is a gentle method?)

Page 3: line 0 and 1: GCs (or GC) instead of GM, because that is not only true for faeces, but a general statement.

Line 39: But see Gorgasser et al (2007), where no effect was seen. I suggest to include that ref (here and/or in the discussion), which indicated that it may depend upon breed or method.

Line 41: Ref 53 is not about FGMs – I suggest to replace it with a most recent review here (Palme, 2019).

Line 42: stress (no plural) or stressors. “decreased/increased” (Past Tense)

Page 4: line 4: why two and three-year-old? You only mention three-year-old elsewhere?

Line 16: “ad libitum” in italics

3.2. – a) and b): I suggest to combine those – there is nothing different in b) – just a few days later. It will also make the figs etc easier to comprehend.

It may be helpful to point to Table S1 already here (or even include a graph in the main text illustrating the timeline of the different situations and sample collections).

Line 35: “quicker”

3.3. – How fresh were collected samples? I think that’s a critical issue, as a recent study by the same research group (why not cite it?) demonstrated.

Line 55 (and elsewhere): I suggest to use “baseline” (adjective) instead of base throughout the ms!

Line 58: I wonder whether those 3 values were normally distributed, and think it may be better to use the median (more representative if there are outliers – and almost identical if not).

Maybe good to mention that the delay time of faecal excretion is about 1 day in horses, and therefore those sampling points were chosen.

Page 5, line 12: The EIA is not directed against 11-oxo-aetiocholanolone-17-CMO (the antibody was raised against that immunogen coupled to BSA). It’s ok just to mention the name of the EIA (maybe include the group of FGMs picked up by this assay).

Line 19-21 (also lines 34 and 35): That corrections sounds odd to me. What were intra-assay CV?

If those were also large, you cannot “correct” with such a factor. By the way, I don’t understand (from the wording) how it was calculated.

Line 27: delete “and mixed in the freezer bag by hand”. You mention this already before – so why another mixing here?

Line 30 and 31: there should be a word space between a number and its dimension (1600 g; 1.2 ml; 3260 g)

Page 6; 3.8. I wonder why inter-observer correlation was calculated for motor laterality (line 8), if only one one did the observations (line 10).

4. a) and onwards. What are the given increases? Taken from mean or median values? Or calculated on an individual basis and from those values the mean/median?

Line 47: I was surprised to read about salivary cortisol here? At least I don’t find it in Fig. 1 (and elsewhere).

Page 7, line 36: Reword: ...to be a good behavioural parameter for the non-invasive evaluation....

Line 41/42: The sentence is hard to understand, please reword.

Line 58: The sample size is always limited, but you probably mean “low”.

References: Please carefully revise them. There are several refs with lacking full information (such as article number, issue or pages; e.g. 21; 29; 59,..)

Ref 24: Delete “Text”

Ref 29: I guess the journal is “Animals” issue/article number?

Ref 53: That is unsuited here – please replace by Palme, 2019.

Fig 1: As mentioned before: please modify “base” to baseline; delete the (b) – as the box is also in yellow you clearly indicate that it is the same situation (maybe rewrite to 1 d, 2 d, 7 d)

Delete the “concentrations” from both y-axes (that’s trivial if you give the dimension).

Any explanation for the extremely high outlier in 2.? I suggest only mentioning it, but redrawing the figure with the y-axes scaled from 0 to 40!

Delete the “trend” (*). It is non-significant, and just makes the figure more complicated.

Fig 2: delete (*) – what is LI? The legends need to be self-explanatory.

References cited above:

Gorgasser, I., Tichy, A., Palme, R. (2007): Faecal cortisol metabolites in Quarter horses during initial training under field conditions. Wien. Tierärztl. Mschr. - Vet. Med. Austria 94, 226-230.

Palme, R. (2019): Non-invasive measurement of glucocorticoids: advances and problems. Physiol. Beh. 199, 229-243.

Author's Response to Decision Letter for (RSOS-191994.R0)

See Appendix A.

Decision letter (RSOS-191994.R1)

28-Jan-2020

Dear Ms Marr,

It is a pleasure to accept your manuscript entitled "Non-invasive stress evaluation in domestic horses (*Equus caballus*): Impact of housing conditions on sensory laterality and immunoglobulin A" in its current form for publication in Royal Society Open Science. The comments of the reviewer(s) who reviewed your manuscript are included at the foot of this letter.

on behalf of Dr Claudia Wascher (Associate Editor) and Kevin Padian (Subject Editor)
openscience@royalsociety.org

Appendix A

Associate Editor's comments (Dr Claudia Wascher):

The presented paper investigates sensory laterality and concentration of faecal immunoglobulin as two potential non-invasive measures of stress in horses. Both reviewers find the study interesting and well presented and they have a number of suggestions to further improve the readability of the manuscript prior to publication.

- Thank you very much. We addressed the suggestion in a point-by-point response to the reviewers comments

Reviewers' Comments to Author:

Reviewer: 1

Comments to the Author(s)

This presents a novel and useful study looking at non-invasive measures of both short-term and long-term stress in horses. This paper is well written, and only requires some small edits to enhance its readability.

- Thank you very much for your valuable comments. They helped to improve the readability of our manuscript a lot.

Page 2

L34 – insert the word “initially” before “significantly shifted to the left...”

- done

L35 – insert “and remained high through initial training” after “individual stabling.”

- done

I know you are probably limited by word count, but I feel it would be useful to include the results of your IgA in the summary.

- This point is well taken. You are right, the abstract is limited by word count. But we still could add 10 words and include short sentence about the IgA results

L45 – change wording “...individual stabling without contact with conspecifics...”

- done

Page 3

L24 – reference 31 deals with differences in right and left hemisphere processing, but in humans, not horses. Please adjust where this reference appears.

- Thank you. We deleted this reference (Royet and Plailly, 2004), as it was not really suitable in this context

L44 - Change to read "...in free-roaming Przewalski and domestic horses.."

L48 – one-sided

L57 – two months (plural)

- L44, L48, L57: Thanks, done.

Page 4

L27-29 - Maybe move the description of the box stalls to the above section 3.1

- done

L35-36 - Can you explain more clearly what responses you expected - ie. what was your hypothesis?

- We added our hypothesis in 3.2. b as follows: *“After one week of individual stabling as described above, the horses were expected to show different coping strategies. Some horses might be quicker to adapt to the novel situation than others than others. Different stress responses were expected including still elevated FGMs, changed motor laterality, decreased faecal IgA (in the case of reactive coping strategies), FGMs at baseline level, unchanged motor laterality, and baseline faecal IgA (in the case of proactive coping strategies). Sensory laterality was expected to return to baseline. The group average would depend on how many animals cope actively or passively.”*

L47 - Please explain this in more detail as "usual practice" will differ among individuals, countries, etc.

- Thank you for this advice. We explained it in more detailed as follows: *“The horses were stabled in single boxes and received regular training (5 times a week) for two months, as is the usual practice in Germany (most horses are stabled in single boxes, training practice is in accordance with the guidelines of the FN).”*

L58 – please rewrite as either "To calculate stable mean base values..." or "To calculate a stable mean base value..."

- done

Page 5

L11 - Please indicate the length of time for centrifugation.

- We added 15 minutes.

L32,33 - Please write out the acronyms EIA and ELISA.

- done

Page 6

L17 - Please explain what the “cs-tool” is.

- Thank you! This was a mistake. cs-tool is the menu item. We used BP-tool that is an Excel Add-In to draw box-plots. We corrected it

L21 - Include the test statistic and p-value for the Shapiro Wilks test.

- done

L47-49 - There is no mention of salivary cortisol analyses in the methodology. Furthermore, Fig 1 shows only values at 24h and 48h after initial training (not immediately or 2h after; nor are these time frames referred to in methodology). Also do not begin the next sentence with a numeral.

- Thank you! We deleted the cortisol values to keep the manuscript simple, as salivary cortisol was investigated elsewhere (Schmidt A, Aurich J, Möstl E, Müller J, Aurich C. 2010 Changes in cortisol release and heart rate and heart rate variability during the initial training of 3-year-old sport horses. *Horm. Behav.* **58**, 628–636. (doi:10.1016/j.yhbeh.2010.06.011)), and our results were in accordance with these results. We forgot to delete this sentence.
- We rewrote the sentence.

Page 7

L17-18 - could it also be that training further developed the "sidedness" of particular horses?

- This point is well taken. Until the initial training (situation c) the horses did not receive any training and, therefore, it can be excluded for test situations a and b. Possibly, after only 1

training session a “sidedness” will not manifest. Therefore, for situation C (initial training), an influence of training on sidedness can also be excluded. For situation c (2 months individual stabling and regular training), possible influences of the training on the sidedness cannot be excluded and need further investigations. We rewrote the sentence in the discussion for a more precise statement: *“Prior to the initial training the observed left shift in the present study was not caused by one sided training [35,37,53], as the horses were already accustomed to being handled from the left in their previous social housing.”*

L41 – months (plural)

- done

L56-57 - However, Squibb et al. (2018) showed that behaviour did not reflect internal shifts in heart rate variability or eye temperature. Perhaps this is due to mature horses used in that study (thus trained to suppress conflict behaviour) versus the untrained horses in this study. Squibb, K., Griffin, K., Favier, R., & Ijichi, C. (2018). Poker Face: Discrepancies in behaviour and affective states in horses during stressful handling procedures. *Applied Animal Behaviour Science*, 202, 34–38. doi:

10.1016/j.applanim.2018.02.003

- This is a very well taken point. This question needs further investigation. The difference between the Squibb study and our study is the kind of stressor in combination with the influence of humans during the test situations. In the Squibb study, horses were encouraged to complete a stressful human-made task (walk under plastic streamers, walk over tarpaulin). As mentioned by the authors, while completing the test situations the stimulus control is a very important factor that influences the behaviour of the horses and can be improved by training. Another influencing factor is the reliability of a horse in responding to the human stimuli that encourage it to complete a task (negative reinforcement). Therefore, the behaviour of the horse during tests like this is influenced by previous training and by the human who is handling the horse during the test and his/her stimuli (timing during reinforcement). In our study horses were also confronted by human-made stressful situations, but there was no further influence by the human that would help indicate solutions (no stimuli, no stimulus control, no reinforcement).

Reviewer: 2

Comments to the Author(s)

The paper submitted by Marr et al (RSOS-191994) reports a “stress” experiment (change of housing; initial training) in horses. The study aims at evaluating two new stress measures (sensory laterality and faecal IgA) by comparing them with two well established ones (FGMs and motor laterality). The latter clearly demonstrated the stressfulness of the situations, which was also found when using sensory laterality (but not faecal IgA). I think the experiment is well planned, the findings are interesting and, although preliminary, the suggested parameters (especially sensory laterality) are worth to be further investigated. Nevertheless there are several things, which to my opinion need to be revised and/or addressed adequately (see my detailed comments below) before the paper is suited for publication.

- Thank you very much for your valuable comments. They helped to improve our manuscript considerably. We hope we have been able to clarify all open questions.

Detailed comments (ordered by line):

Page 2, line 32 (and elsewhere) I suggest adding the “s” for the plural: FGMs – in some cases the singular is needed (e.g. faecal glucocorticoid metabolite concentrations – FGM)

- Thank you! We checked the whole manuscript.

Line 37: modify to “level, as a behavioural parameter that...(I may be mistaken, but think that observing behaviour is always non-invasive, or?)

- done

Abstract: I miss a word about unaffected IgA levels.

- We added information about IgA results

Line 46: stress responses (plural)

Line 49: better use “stressors” instead of “stress” (two times)

Line 57: Minimal to non-invasive methods ,..., such as glucocorticoids in saliva or their metabolites in faeces... (what is a gentle method?)

- Line 46, 49, 57: done and corrected

Page 3: line 0 and 1: GCs (or GC) instead of GM, because that is not only true for faeces, but a general statement.

- done

Line 39: But see Gorgasser et al (2007), where no effect was seen. I suggest to include that ref (here and/or in the discussion), which indicated that it may depend upon breed or method.

- Thank you. We added the Gorgasser et al. study [52] to the introduction: *“Previous studies have demonstrated that stressors such as initial training [51] and changes from social housing to individual housing (which restricts movement and social interaction), increase the concentration of cortisol both in horses’ saliva and of GMs in the faeces [1,2], whereas the initial training increased cortisol concentrations in horses’ saliva [51] but not the GM concentrations in faeces [52].”*

Line 41: Ref 53 is not about FGMs – I suggest to replace it with a most recent review here (Palme, 2019).

- Thank you. We deleted the reference and added Palme (2019) [76] to the discussion: *“In stress situations similar to those in the present study behavioural parameters have been shown to be potentially more indicative in situations where stress hormones did not show any change, for example after changed housing conditions [75]. Nonetheless, in addition to changes in laterality the evaluation of other stress parameters it is also recommended for a reliable assessment of stress responses [76].”*

Line 42: stress (no plural) or stressors. “decreased/increased” (Past Tense)

- done

Page 4: line 4: why two and three-year-old? You only mention three-year-old elsewhere?

- The horses were between 2 and 3 year-old. We changed it to three-year-old and added the year foaled for more precise information.

Line 16: “ad libitum” in italics

- done

3.2. – a) and b): I suggest to combine those – there is nothing different in b) – just a few days later. It will also make the figs etc easier to comprehend.

- We would like to keep it as it is, to allow an easier comparison of the physiological and behavioural parameters. One week of individual stabling was expected to elicit long-term stress response and the change from group to individual housing was expected to elicit short-term stress response. Therefore we chose different labels.

It may be helpful to point to Table S1 already here (or even include a graph in the main text illustrating the timeline of the different situations and sample collections).

- Thank you for this advice. We changed the first sentence of 3.2. and point to Table S1

Line 35: “quicker”

- done

3.3. – How fresh were collected samples? I think that’s a critical issue, as a recent study by the same research group (why not cite it?) demonstrated.

- Please see 3.4. There we mentioned how fresh the collected samples were. Now, we added more detailed information: *“Fresh faecal samples were collected from the centre of the pile with a freezer bag between one and two minutes after defecation and kneaded for one to two minutes to ensure an equal distribution of IgA and GMs. They were then kept on ice in a cool box until they were frozen in the laboratory at -20°C to avoid a decline in FGM concentration through bacterial decay as demonstrated elsewhere [58].”*

Line 55 (and elsewhere): I suggest to use “baseline” (adjective) instead of base throughout the ms!

- done

Line 58: I wonder whether those 3 values were normally distributed, and think it may be better to use the median (more representative if there are outliers – and almost identical if not).

- A good point. Before we calculated the mean baseline value for each horse, we tested whether the 3 measurements differ by using the Friedman-test. There were no significant differences between the 3 measurements (FGM: chi-squared = 4.6, df = 2, p = 0.103; IgA: chi-squared = 0.2, df = 2, p = 0.0.91). Therefore, we decided to calculate a mean baseline value to include “normal” variations between the days.

Maybe good to mention that the delay time of faecal excretion is about 1 day in horses, and therefore those sampling points were chosen.

- Thank you for this valuable advice. We *added the information under 3.3.: “As horses’ intestinal passage takes about 24 hours [55] faecal samples were taken during each test situation as follows:”*

Page 5, line 12: The EIA is not directed against 11-oxo-aetiocholanolone-17-CMO (the antibody was raised against that immunogen coupled to BSA). It’s ok just to mention the name of the EIA (maybe include the group of FGMs picked up by this assay).

- We rewrote the sentence: *“As validated and described for horses elsewhere [59], the diluted supernatant (in assay buffer) was analysed by an 11-oxo-aetiocholanolone enzyme immunoassay measuring 11,17-dioxoandrostanes.”*

Line 19-21 (also lines 34 and 35): That corrections sounds odd to me. What were intra-assay CV? If those were also large, you cannot “correct” with such a factor. By the way, I don’t understand (from the wording) how it was calculated.

- The intra-assay CV’s of all samples were below 10%. All plates had the same control sample $CS_{plate\ x}$. A mean value of the control samples from all plates was calculated CS_{mean} . The

correction factor $CF_{\text{plate } x}$ for Plate x was calculated as follows: $CS_{\text{mean}}/CS_{\text{plate } x}$. And the values of the test situations from plate x were then multiplied by $CF_{\text{plate } x}$. We hope it is clearer now.

Line 27: delete “and mixed in the freezer bag by hand”. You mention this already before – so why another mixing here?

Line 30 and 31: there should be a word space between a number and its dimension (1600 g; 1.2 ml; 3260 g)

- Line 27, 30, 31: done

Page 6; 3.8. I wonder why inter-observer correlation was calculated for motor laterality (line 8), if only one one did the observations (line 10).

- “Before the experiment started, an inter-observer agreement between the two experimenters was tested for the assessment of sensory and motor laterality and a Cohen’s Kappa coefficient was calculated (sensory laterality: $\kappa = 0.75$, motor laterality: $\kappa = 0.93$).” (3.8. Experimenters) Both experimenters observed horses at the same time to calculate a Cohen’s Kappa coefficient before the experimenter started. During the experiment only one of the experimenters observed the horses, while the other experimenter collected samples etc.

4. a) and onwards. What are the given increases? Taken from mean or median values? Or calculated on an individual basis and from those values the mean/median?

- Significant increases/changes mean an increase/change from the calculated mean baseline values of the horses to the values of the particular test situation taking the repeated measurement into consideration. We calculated the difference for each horse and test situation between its mean baseline value and the value of the particular test situation (repeated measurement design of the study). By using a GLM we tested whether these differences were significant different from 0 (which means no change) for each test situation.

Line 47: I was surprised to read about salivary cortisol here? At least I don’t find it in Fig. 1 (and elsewhere).

- Thank you! We deleted the cortisol values to keep the manuscript simple, as salivary cortisol was investigated elsewhere (Schmidt A, Aurich J, Möstl E, Müller J, Aurich C. 2010 Changes in cortisol release and heart rate and heart rate variability during the initial training of 3-year-old sport horses. *Horm. Behav.* **58**, 628–636. (doi:10.1016/j.yhbeh.2010.06.011)), and our results were in accordance with these results. We forgot to delete this sentence.

Page 7, line 36: Reword: ...to be a good behavioural parameter for the non-invasive evaluation....

- done

Line 41/42: The sentence is hard to understand, please reword.

- We rewrote the sentence and hope it is more readable now: “*Besides implicating animal welfare issues [25], a left shift in sensory and motor laterality indicating an increased information processing by the right hemisphere could indicate training and handling issues.*”

Line 58: The sample size is always limited, but you probably mean “low”.

- Thanks! Corrected.

References: Please carefully revise them. There are several refs with lacking full information (such as article number, issue or pages; e.g. 21; 29; 59,..)

Ref 24: Delete “Text”

Ref 29: I guess the journal is “Animals” issue/article number?

Ref 53: That is unsuited here – please replace by Palme, 2019.

- Thank you. We revised them.

Fig 1: As mentioned before: please modify “base” to baseline; delete the (b) – as the box is also in yellow you clearly indicate that it is the same situation (maybe rewrite to 1 d, 2 d, 7 d)

Delete the “concentrations” from both y-axes (that’s trivial if you give the dimension).

- We modified base to baseline and deleted the “concentrations”
- We would like to keep the labeling as it is, to allow an easier comparison of the physiological and behavioural parameters. One week of individual stabling was expected to elicit long-term stress response and the change from group to individual housing was expected to elicit short-term stress response. Therefore we chose different labels.

Any explanation for the extremely high outlier in 2.? I suggest only mentioning it, but redrawing the figure with the y-axes scaled from 0 to 40!

- The extremely high outlier at 24 h after the change of housing conditions is the same horse that is an outlier also in FGM in the same test situation. According to its behaviour this horse seemed to be extremely stressed by the novel situation. We redrew the y-axes and mentioned the outlier in the figure caption.

Delete the “trend” (*). It is non-significant, and just makes the figure more complicated.

- We deleted the trend

Fig 2: delete (*) – what is LI? The legends need to be self-explanatory.

- We deleted the trend and changed LI to “laterality index”.